# RIGOROUS AGENT EVALUATION: AN ADVERSARIAL APPROACH TO UNCOVER CATASTROPHIC FAILURES

**Jonathan Uesato**\*, **Ananya Kumar**\*, **Csaba Szepesvari**\*, **Tom Erez, Avraham Ruderman**
**Keith Anderson, Krishmamurthy (Dj) Dvijotham, Nicolas Heess, Pushmeet Kohli**
DeepMind, London, UK
{juesato, ananyak, szepi, pushmeet}@google.com

## ABSTRACT

This paper addresses the problem of evaluating learning systems in safety critical domains such as autonomous driving, where failures can have catastrophic consequences. We focus on two problems: searching for scenarios when learned agents fail and assessing their probability of failure. The standard method for agent evaluation in reinforcement learning, Vanilla Monte Carlo, can miss failures entirely, leading to the deployment of unsafe agents. We demonstrate this is an issue for current agents, where even matching the compute used for training is sometimes insufficient for evaluation. To address this shortcoming, we draw upon the rare event probability estimation literature and propose an *adversarial evaluation* approach. Our approach focuses evaluation on adversarially chosen situations, while still providing unbiased estimates of failure probabilities. The key difficulty is in identifying these adversarial situations – since failures are rare there is little signal to drive optimization. To solve this we propose a *continuation approach* that learns failure modes in related but less robust agents. Our approach also allows reuse of data already collected for training the agent. We demonstrate the efficacy of adversarial evaluation on two standard domains: humanoid control and simulated driving. Experimental results show that our methods can find catastrophic failures and estimate failures rates of agents multiple orders of magnitude faster than standard evaluation schemes, in minutes to hours rather than days.

## 1 INTRODUCTION

How can we ensure machine learning systems do not make catastrophic mistakes? While machine learning systems have shown impressive results across a variety of domains (Krizhevsky et al., 2012; Mnih et al., 2015; Silver et al., 2017), they may also fail badly on particular inputs, often in unexpected ways (Szegedy et al., 2013). As we start deploying these systems, it is important that we can reliably evaluate the risk of failure. This is particularly important for safety critical domains like autonomous driving where the negative consequences of a single mistake can overwhelm the positive benefits accrued during typical operation of the system.

**Limitations of random testing.** The key problem we highlight is that for standard statistical evaluation, attaining confidence that the failure rate of a policy is below $\epsilon$ requires at least $1/\epsilon$ episodes. We now informally summarize this point, which we discuss further in Appendix A. For concreteness, consider a self-driving car company that decides that the cost of a single accident where the car is at fault outweighs the benefits of 100 million miles of faultless operation. The standard approach in machine learning is to estimate the expected return via i.i.d. samples from the data distribution (frequently a test set). For tightly bounded returns, the sample estimate is guaranteed to quickly converge to the true expectation. However, with catastrophic failures, this may be prohibitively inefficient. In our current example, any policy with a failure probability greater than $\epsilon = 10^{-8}$ per mile has negative expected return. In other words, it would be better to not deploy the car. However, to achieve reasonable confidence that the car crashes with probability below $1e$–$8$, the manufacturer would need to test-drive the car for at least $1e8$ miles, which may be prohibitively expensive.

---

\*Equal Contribution

**Our Contributions.** To overcome the above-mentioned problems, we develop a novel *adversarial evaluation* approach. The central motivation behind our algorithmic choices is the fact that real-world evaluation is typically dominated by the cost of running the agent in the real-world and/or human supervision. In the self-driving car example, both issues are present: testing requires both operating a physical car, and a human test driver behind the wheel. The overarching idea is thus to screen out situations that are unlikely to be problematic, and focus evaluation on the most difficult situations. The difficulty arises in identifying these situations – since failures are rare, there is little signal to drive optimization. To address this problem, we introduce a *continuation approach* to learning a *failure probability predictor* (AVF), which estimates the probability the agent fails given some initial conditions. The idea is to leverage data from less robust agents, which fail more frequently, to provide a stronger learning signal. In our implementation, this also allows the algorithm to reuse data gathered for training the agent, saving time and resources during evaluation. We note that adversarial testing is a well-established idea (see Section 5), but typically requires either a dense optimization signal or expert domain knowledge. We avoid these stumbling blocks by relying on the learned AVF, which guides the adversarially acting evaluator.

We look at two settings where the AVF can be used. In the simplest setting, *failure search*, the problem is to efficiently find inputs (initial conditions) that cause failures (Section 2.1). This task has several uses. First, an adversary that solves this task efficiently allows one to identify and debug potentially unsafe policies. Second, as has been done previously in the supervised learning literature, efficient adversaries can be used for adversarial training, by folding the states causing failures back into the training algorithm (Madry et al., 2017). The second setting, *risk estimation*, is the problem of efficiently estimating the failure probability of an agent (Section 2.2), which also has a simple application to efficiently selecting the most reliable agent from a finite set (Section 4.3).

Empirically, we demonstrate dramatic improvements in efficiency through adversarial testing on two domains (simulated driving and humanoid locomotion). In summary, we present 3 key contributions:

1. We empirically demonstrate the limitations of random testing. We observe that with random testing, the cost of reliably obtaining even a single adversarial input can exceed the entire cost of training. Further, reliably estimating risk can exceed training costs.

2. We describe a continuation approach for learning *failure probability predictors* even when failures are rare. We develop algorithms applying failure probability predictors to failure search, risk estimation, and model selection.

3. We extensively evaluate our method on simulated driving and humanoid locomotion domains. Using adversarial evaluation, we find failures with 198 and 3100 times fewer samples respectively. On Humanoid, we bring the cost of reliable risk estimation down from greater than the cost of training to a practical budget.

## 2 PROBLEM FORMULATION

We first introduce our notation. Recall that we are interested in reliability assessment of a trained agent. We assume that the experimenter who performs the reliability assessment can run an *experiment* (equivalently, a rollout or episode) with the trained agent given some initial condition $x \in \mathcal{X}$, the outcome of which is a random failure indicator $C = c(x, Z)$ where $Z \sim P_Z$ for some probability distribution $P_Z$ over some set $\mathcal{Z}$ and where $c : \mathcal{X} \times \mathcal{Z} \rightarrow \{0, 1\}$. In particular, $C$ is binary and $C = 1$ indicates a "catastrophic failure". The interpretation of $Z$ is that it collects all sources of randomness due to the agent and the environment. Unlike the case of $x$, $Z$ is neither observed, nor controllable by the experimenter. We are interested in the agent's performance on the environment distribution over initial conditions $X \sim P_X$, which we assume can be sampled quickly.[1]

In the real-world, evaluating $c$ requires running the agent in the real-world and/or human supervision, which generally dominate the cost of evaluating neural networks. It is thus assumed that evaluating $c$ on the pair $(x, Z)$ is costly. In Section 4.1, we show that even on relatively cheap simulated environments, the cost of simulating environments dominates costs of evaluating neural networks.

---

[1] To minimize jargon, we omit standard conditions on the spaces and functions that permit the use of the language of probabilities.

## 2.1 FAILURE SEARCH

The objective of *failure search* is to find a catastrophic failure of the agent. Algorithms that search for failures, which we will call *adversaries*, can specify initial conditions $x$, observe the outcome $C$ of running the agent on $x$ and return as soon as $C = 1$. Thus, for each round $t$, the adversary can choose $X_t \sim P_t(\cdot | X_1, \ldots, X_{t-1})$, observe $C_t = c(X_t, Z_t)$, and return if $C_t = 1$. Here, $Z_t \sim P_Z$, independently of the history $(X_1, \ldots, X_t, Z_1, \ldots, Z_{t-1})$. The *naive adversary* evaluates the agent on samples from $P_X$ until observing a failure. We assess adversaries by either the expected number of episodes, or the expected time elapsed, until returning a failure case.

To design a more efficient adversary, the experimenter is allowed to collect *historical data* of the form $(X'_1, \theta_1, C'_1), \ldots, (X'_n, \theta_n, C'_n)$ *while they are training their agent*. Here, the initial condition $X'_t$ and $\theta_t \in \Theta$ are chosen by the training procedure, where $\theta_t$ encodes information about the agent used in round $t$ to generate the observation $C'_t = c(X'_t, Z'_t)$.

## 2.2 RISK ESTIMATION

As before, let $P_X$ denote the distribution over initial states $\mathcal{X}$ defined by the environment. The *failure probability* of the trained agent is

$$p = \mathbb{E}[c(X, Z)],$$

where $X \sim P_X$ and $Z \sim P_Z$ independently. A typical goal is to estimate $p$ up to a fixed *relative accuracy* with high probability. Given some $\rho > 1$, $\delta \in (0, 1)$, an algorithm is said to be $(\rho, \delta)$ correct if the estimate $\hat{P}$ produced belongs to the $[p/\rho, p\rho]$ interval with probability at least $1 - \delta$. When $\hat{P}$ belongs to the said interval, we say that it is a $\rho$-approximation of $p$.

As before, the algorithm has access to historical data and can quickly sample from $P_X$. The naive or *vanilla Monte Carlo* (VMC) estimator samples from $P_X$ and returns the average of the observed outcomes. We assess estimators by the total number of experiments required.

# 3 APPROACH

In this section we describe our approach to the two problems outlined above. A common feature of our proposed solutions is that they estimate the *failure probability predictor* (AVF) $f_* : \mathcal{X} \to [0, 1]$ that returns the probability of failure given a initial condition (equivalently, the value function of an adversary which receives reward 1 for failures, without discounting):

$$f_*(x) = \mathbb{P}(c(x, Z) = 1), \qquad x \in \mathcal{X},$$

where $Z \sim P_Z$. Our *continuation* approach to learning an approximation $f \approx f_*$, $f : \mathcal{X} \to [0, 1]$ is described in Section 3.3. $f$ will be chosen so that the cost of evaluating $f$ is negligible compared to running an experiment. The idea is to use $f$ to save on the cost of experimentation. Our solutions build on the certainty equivalence approach (Turnovsky, 1976): First, we describe how $f_*$ (if available) could be leveraged. In cases of certainty equivalence, we would substitute $f$ for $f_*$. We then introduce additional heuristics to reduce sensitivity to the mismatch between $f$ and $f_*$.

## 3.1 FAILURE SEARCH

When $f_*$ is known, the optimal adversary has a particularly simple form (proof left to the reader):

**Proposition 3.1.** *The adversary that minimizes the expected number of rounds until failure evaluates the agent repeatedly on an instance $x_* \in \mathcal{X}$ that maximizes $f_* : \mathcal{X} \to [0, 1]$.*

Having access to $f \approx f_*$, a natural approach is to evaluate the agent on $\arg\max_x f(x)$. There are two problems with this: *(i)* a maximizer may be hard to find and *(ii)* there is no guarantee that the maximizer of $f$ will give a point where $f_*$ is large. Rather than always select the global maximizer of $f$, one way to increase the robustness of this procedure is to sample diverse initial conditions. We adopt a simple procedure: sample $n$ initial conditions from $P_X$, pick the initial condition from this set where $f$ is the largest, and run an experiment from the found initial condition. We repeat this process with new sampled initial conditions until we find a catastrophic failure. The pseudocode is included in Appendix C as Algorithm 2 (AVF Adversary).

## 3.2 RISK ESTIMATION USING AVFS

The failure probability estimation method uses importance sampling (IS) (e.g., Section 4.2, Bucklew 2004) where the distribution $P_X$ is replaced by a *proposal distribution $Q$*. For $t \in [n]$, the proposal distribution is used to generate random initial condition $X_t \sim Q$, then an experiment is performed to generate $C_t = c(X_t, Z_t)$. The failure probability $p$ is estimated using the sample mean

$$\hat{P} = \frac{1}{n} \sum_{t=1}^{n} W_t \, c(X_t, Z_t),$$

where $W_t = \frac{p_X(X_t)}{q(X_t)}$ is the *importance weight* of the $t$-th sample. Here, $p_X$ denotes the density of $P_X$ and $q$ denotes the density of $Q$.[2] Let $U_t = W_t c(X_t, Z_t)$. As is well-known, under the condition that $p_X(x)f^*(x) = 0$ whenever $q(x) = 0$, we have that $\mathbb{E}[U_t] = p$, and hence $\mathbb{E}[\hat{P}] = p$. Given that $\hat{P}$ is unbiased for any proposal distribution, one natural objective is to choose a proposal distribution which minimizes the variance of the estimator $\hat{P}$.

**Proposition 3.2.** *For $f : \mathcal{X} \to [0,1]$ such that $P_X(f^{1/2}) := \int f^{1/2}(x')P_X(dx') > 0$ let $Q_f$ be defined as the distribution over $\mathcal{X}$ whose density $q_f$ is*

$$q_f(x) = \frac{f^{1/2}(x)p_X(x)}{P_X(f^{1/2})} \, .$$

*Then, the variance minimizing proposal distribution $Q^*$ is $Q_{f_*}$.*

The (standard) proof of this and the next proposition can be found in Appendix B. Note that from the definition of $Q_f$ it follows that $\frac{p_X(x)}{q_f(x)} = f^{-1/2}(x)P_X(f^{1/2})$ when $q_f(x) > 0$.

The above result motivates us to suggest using the distribution $Q_f$ as the proposal distribution of an IS estimator. Note that the choice of $f$ (as long as it is bounded away from zero) only influences the efficiency of the method, but not its correctness. It remains to specify how to sample from $Q_f$ and how to calculate the importance weights. For sampling from $Q_f$, we propose to use the *rejection sampling method* (Section 1.2.2, Bucklew 2004) with the proposal distribution chosen to be $P_X$: First, $X \sim P_X$ is chosen, which is accepted by probability $f^{1/2}(X)$. This is repeated until a sample is accepted:

**Proposition 3.3.** *Rejection sampling as described produces a sample from $Q_f$.*

To increase the robustness of the sampling procedure against errors introduced by $f \neq f_*$, we introduce a "hyperparameter" $\alpha > 0$ so that $q_f$ is redefined to be proportional to $f^\alpha$. Note that $\alpha = 1/2$ is what our previous result suggests. However, if $f$ is overestimated, a larger value of $\alpha$ may work better, while if $f$ is underestimated, a smaller value of $\alpha$ may work better. The pseudocode of the full procedure is given as Algorithm 1.

## 3.3 A CONTINUATION APPROACH TO LEARNING AVFS

Recall that we wish to learn an approximation $f$ to $f^*$, where $f^*$ is the true failure probability predictor for an agent $A$. The classical approach is to first obtain a dataset $(X_1, C_1), \ldots, (X_n, C_n)$ by evaluating the agent of interest on initial states $X \sim P_X$. We could then fit a softmax classifier $f$ to this data with a standard cross-entropy loss.

The classical approach does not work well for our setup, because failures are rare and the failure signal is binary. For example, in the humanoid domain, the agent of interest fails once every 110k episodes, after it was trained for 300k episodes. To learn a classifier $f$ we would need to see many failure cases, so we would need significantly more than 300k episodes to learn a reasonable $f$.

To solve this problem, we propose a *continuation approach* to learning AVFs, where we learn $f$ from a family of related agents that fail more often. The increased failure rates of these agents can provide an essential boost to the 'signal' in the learning process.

---

[2]Here, we implicitly assume that these measures have a density with respect to a common measure. As it turns out, this is not a limiting assumption, but this goes beyond the scope of the present article.

---

**Algorithm 1** AVF-guided risk estimator (AVF estimator)

---

**Input:** AVF $f$, budget $T \in \mathbb{N}$ and tuning parameter $\alpha > 0$
**Returns:** Failure probability estimate $\hat{P}$
Initialize $S \leftarrow 0$
**for** $t = 1$ **to** $T$ **do**
   **repeat**
      Sample proposal instance $X_t \sim P_X$
      Accept $X_t$ with probability $f^\alpha(X_t)$
   **until** accepting initial condition $X_t$
   Evaluate the agent on $X_t$ and observe $C_t = c(X_t, Z_t)$
   $S \leftarrow S + C_t / f^\alpha(X_t)$
**end for**
Compute normalization constant $Z \leftarrow \frac{1}{m} \sum_{i=1}^m f^\alpha(X_i')$ for $X_i' \sim P_X, m \gg T$
**return** $\hat{P} \leftarrow ZS/T$

---

The particular problem we consider, agent evaluation, has additional structure that we can leverage. Typically, agents earlier on in training fail more often but in related ways, to the final agent. So we propose learning $f$ from agents that were seen earlier on in training. This has an added computational benefit – we do not need to gather additional data at evaluation time. Concretely, we collect data during training of the form $(X_1', \theta_1, C_1'), \ldots, (X_n', \theta_n, C_n')$ where $X_t'$ is the initial condition in iteration $t$ of training, and $\theta_t$ characterizes essential features of the agent's policy used to collect the failure indicator $C_t'$. We then train a network to predict $\mathbb{P}(C_t' = 1 | X_t', \theta_t)$ at input $(X_t', \theta_t)$. In the simplest implementation, $\theta_t$ merely encodes the training iteration $t$. In many common off-policy RL methods, additional stochasticity is used in the policy at training time to encourage exploration. In these cases, we also use $\theta_t$ to encode the degree of stochasticity in the policy.

The continuation approach does make a key assumption – that agents we train on fail in related ways to the final agent. We provide a simple toy example to discuss in what ways we rely on this, and in what ways we do not.

**Example.** Suppose the true AVF $f_*$ factorizes as $f_*(x, \theta) = g_*(x) h_*(\theta)$. Suppose we have two agents described by $\theta_1, \theta_2$, such that the latter agent is significantly more robust, i.e., $h^*(\theta_1)/h^*(\theta_2) = r \gg 1$. Then, if learning requires a fixed number of positive examples to reach a particular accuracy, the continuation approach learns $g$, the state-dependent component, $r$ times faster than the naive approach, since it receives $r$ times as many positive examples.

On the other hand, accurately learning $h(\theta_2)$ is difficult, since positive examples on $\theta_2$ are rare. However, even when $h$ is learned inaccurately, this AVF is still useful for both failure search and risk estimation. Concretely, if two AVFs $f_1(x, \theta) = g(x)h_1(\theta), f_2(x, \theta) = g(x)h_2(\theta)$ differ only in $h$, then *both the adversaries and estimators induced by $f_1$ and $f_2$ are equivalent*.

**Discussion.** Of course, this strict factorization does not hold exactly in practice. However, we use this example to illustrate two points. First, for both failure search and risk estimation, it is the shape of the learned AVF $f(\theta, s)$ which matters more than its magnitude. Second, it captures the intuition that the AVF may learn the underlying structure of difficult states, i.e. $g(s)$, on policies where positive examples are less scarce. Further, using flexible parameterizations for $f$, such as neural networks, may even allow $f$ to represent interactions between $\theta$ and $s$, provided there is sufficient data to learn these regularities. Of course, if failure modes of the test policy of interest look nothing like the failure modes observed in the related agents, the continuation approach is insufficient. However, we show that in the domains we study, this approach works quite well.

## 4 EXPERIMENTS

We run experiments on two standard reinforcement learning domains, which we describe briefly here. Full details on the environments and agent training procedures are provided in Appendix E.

In the *Driving* domain, the agent controls a car in the TORCS simulator (Wymann et al., 2000) and is rewarded for forward progress without crashing. Initial conditions are defined by the shape of the track, which is sampled from a procedurally defined distribution $P_X$. We define a failure as

| Domain | AVF Cost | VMC Cost | PR Cost | Acceleration Factor |
|---|---|---|---|---|
| Driving | 3/5/11 | 200/1000/2700 | —* | 65/198/250 |
| Humanoid | 19/33/56 | 60K/110K/180K | 9K/10K/220K* | 2100/3100/3800 |

Table 1: **Failure search.** We show cost for different adversaries to find an adversarial problem instance, measured by the expected number of episodes. Each column reports the min, median, and max over evaluations of 5 separate agents. In the median case, the AVF adversary finds adversarial inputs with 198x fewer episodes than random testing on Driving and 3100x fewer on Humanoid (Acceleration Factor).

any collision with a wall. We use an on-policy actor-critic agent as in Espeholt et al. (2018), using environment settings from Mnih et al. (2016) which reproduces the original results on TORCS.

In the *Humanoid* domain, the agent controls a 21-DoF humanoid body in the MuJoCo simulator (Todorov et al., 2012; Tassa et al., 2018), and is rewarded for standing without falling. The initial condition of an experiment is defined by a standing joint configuration, which is sampled from a fixed distribution. We define a catastrophic failure as any state where the humanoid has fallen, i.e. the head height is below a fixed threshold. We use an off-policy distributed distributional deterministic policy gradient (D4PG) agent, following hyperparameters from Barth-Maron et al. (2018) which reproduces the original results on the humanoid tasks.

### 4.1 FAILURE SEARCH

We first compare three adversaries by the number of episodes required to find an initial condition that results in a trajectory that ends with a failure. Our purpose is to illustrate two key contributions. First, a naive evaluation, even when using the same number of episodes as training the agent, can lead to a false sense of safety by failing to detect any catastrophic failures. Second, adversarial testing addresses this issue, by dramatically reducing the cost of finding failures. The adversaries evaluated are the naive (VMC) and the AVF adversaries introduced in Section 3.1, as well as an adversary that we call the *prioritized replay* (PR) adversary. This latter adversary runs experiments starting from all initial conditions which led to failures during training, most recent first. This provides a simple and natural alternative to the naive adversary. Additional details on all adversaries are provided in Appendix E. The results are summarized in Table 1.

**Discussion of AVF adversary.** The AVF adversary is multiple orders of magnitude more efficient than the random adversary. In particular, we note that on Humanoid, for the VMC adversary to have over a 95% chance of detecting a single failure, we would require over $300,000$ episodes, exceeding the cost of training, which used less than $300,000$ episodes[3]. In practice, evaluation is often run for much less time than training, and in these cases, the naive approach would very often lead to the mistaken impression that such failure modes do not exist.

We observe similar improvements in wall-clock time. On the Driving domain, in the median case, finding a failure requires 2.7 hours in expectation for the random adversary, compared to 1.1 minutes for the AVF adversary. Even including the model training time of 5 minutes, the AVF adversary is 27 times faster. Similarly, on the Humanoid domain, the random adversary requires 3 days (77 hours) in expectation, compared to 6 minutes for the AVF adversary, amounting to a 61-fold speedup after including AVF training time of 70 minutes. These numbers underscore the point that even in relatively cheap, simulated environments, the cost of environment interactions dominate learning and inference costs of the AVF.

**Discussion of Prioritized Replay adversary.** The PR adversary often provides much better efficiency than random testing, with minimal implementational overhead. The main limitation is that in some cases, the prioritized replay adversary may *never* detect failures, even if they exist. In particular, an agent may learn to handle the particular problem instances on which it was trained, without eliminating all possible failures. In these cases (this occurred once on Humanoid), we fall back to the VMC adversary after trying all the problem instances which caused failures at training. On the

---

[3]The number of failures is distributed $Bin(N,p)$ and in the median case, for $p = 1/110,000$, $N = 300,000$, we have $(1-p)^N > 0.05$.

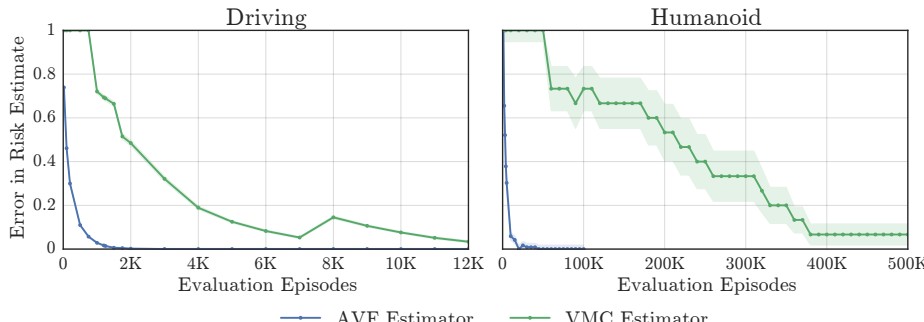

Figure 1: The figure shows the reliability of the AVF and VMC estimators. The $x$ axis shows the number of evaluation episodes, while the $y$ axis shows the probability that $\hat{P}$ does not belong to the interval $(p/3, 3p)$. This probability is obtained by repeating the evaluation multiple times and plotting the fraction of these runs when the estimate is outside of the target interval. The AVF estimator reliably approaches ground truth dramatically faster than the VMC estimator. We show error bars of 2 standard errors, which are difficult to see on the Driving domain, since we can afford to run sufficiently many trials that standard errors are near zero.

Driving domain, because we use a form of adversarial training for the agent, very few ($< 20$) of the training problem instances have support under $P_X$. Since none of these resulted in failures, the PR adversary would immediately fall back to the VMC adversary.

**Comparison to classical approaches.** One question is why we do not compare to more classical techniques from the rare events literature, such as cross-entropy method or subset simulation. Because we optimize a binary failure signal, rather than a continuous score as in the classical setting, these approaches would not work well. For example, in Humanoid, the final agent fails once every 110k episodes, and was trained for 300k episodes. If we used cross-entropy method on the final agent, we would need significantly more than 300k episodes of data to fit a good proposal distribution. The continuation approach is essential - obtaining enough failures to fit a useful proposal distribution requires learning from a family of weaker agents.

## 4.2   RISK ESTIMATION

We now compare approaches for risk estimation. Again, the purpose is to illustrate the claim that a naive approach may often be too inefficient to give rise to non-vacuous quantitative answers with reasonable resources, while better alternatives can deliver such answers. In particular, we compare the AVF estimator of Section 3.2 to the vanilla Monte Carlo estimator (VMC). For comparison purposes, we run each estimator multiple times and report the fraction of cases when the obtained estimates fail to fall in the $(p/3, 3p)$ interval, where $p$ is the failure probability to be estimated. This provides a fairly accurate estimate of the probability of $\hat{P}$ failing to be a $\rho = 3$-approximation of $p$.[4] Results are summarized in Fig. 1. In Appendix E.3 we include plots for other choices of $\rho$, to show that the results are not very sensitive to $\rho$.

**Discussion.** At the confidence level $1 - \delta = 0.95$, on the Driving domain, the AVF estimator needs only 750 experiments to achieve a 3-approximation, while the VMC estimator needs $11,000$ experiments for the same. This means that here the AVF estimator requires 14 times less environment interaction. Similarly, on the Humanoid domain, at the confidence level $1 - \delta = 0.95$, the AVF estimator requires $15,000$ experiments to achieve a 3-approximation, while the VMC estimator requires $5.1e5$ experiments, a 34-fold improvement.

We note that these numbers demonstrate that the AVF also has good *coverage*, i.e. it samples from most possible failures, since if a large number of failure conditions were vastly undersampled, the variance of the AVF estimator would explode due to the importance weight term. Coverage and efficiency are complementary. While failure search merely requires the adversary to identify a single failure, efficient risk estimation requires the adversary to sample from most possible failures.

---

[4]Here, $p$ was measured separately by running the VMC estimator for $5e6$ episodes on Driving and $2e7$ episodes on Humanoid, so that 2 standard errors lies within 5% relative error on Driving, and 20% on Humanoid.

### 4.3 Application to identifying more reliable agents

The improved efficiency of the AVF estimator has many benefits. One application is to identify the most reliable agent from a fixed finite set.

For this illustration, we compare 50 Humanoid agents, spaced evenly over the course of training (excluding the beginning of training, when the agent has a high failure rate). We compare the choices made when the failure probability estimation is performed with the VMC estimator versus the AVF estimator. The selected policy at each point in time is the policy with the lowest estimated risk. In the event of a tie (e.g., among several policies with no failures), we report the expected failure probability when selecting from the tied agents randomly.

Fig. 2 summarizes the results. When using VMC on the median agent, 36 out of 50 policies have never failed by the end of the evaluation process. Thus, VMC was not able to rank 36 out of the 50 policies and the experimenter would be forced to choose one of these with no further information about their reliability. On the other hand, the AVF estimator quickly produces failures for the weak policies, and is able to

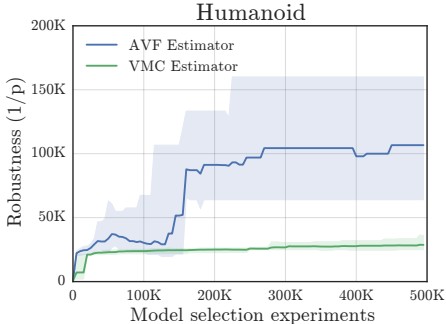

Figure 2: **Model selection**: We plot the expected number of episodes until failure, i.e. $1/p$ where $p$ is the probability of failure, for the best policy selected by the AVF vs. VMC estimators. The error bars show the min/max over 5 random seeds, while the solid lines correspond to the averaged failure probability. The VMC estimator largely maintains a uniform distribution over many of the policies, whereas the AVF estimator quickly eliminates the worst policies.

select a policy which is over 3 times more robust. This can be viewed as an extremely naive form of adversarial training, where the model is selected from a discrete set of options rather than a high-dimensional parameter space. We use this example to highlight the recurring theme that random testing can be highly inefficient – failing to identify the best policies even after $500,000$ episodes (longer than training time) – while adversarial testing greatly improves efficiency.

### 4.4 Applicability and Robustness

We conclude this section with practical considerations regarding the use of this approach in settings beyond the domains considered here, including the real world.

**Maintaining high coverage.** A concern is that if the AVF underestimates the failure probability of certain initial conditions, the risk estimator will have high variance, and with limited samples, often underestimate the true failure probability. Our main point is that in many settings, VMC has little chance of revealing any failures with a practical budget, and so using any form of optimized adversary is a large improvement, but we also acknowledge other factors which mitigate this issue.

First, because the AVF is trained on weaker agents, it typically over-estimates failure probabilities. Second, in Humanoid, where failures are particularly rare, we use a simplified Differentiable Neural Dictionary (DND) (Pritzel et al., 2017) described in Appendix E.1. A DND is a kNN classifier in feature space, but uses a learned pseudo-count, which causes it to output higher failure probabilities when the query point is far in feature space from training points.

**Efficiency lower bounds.** Further, we can ensure our method never does more than 2 times worse than VMC. To do so, we run both the VMC and AVF estimators in parallel. If VMC finds at least several failures, then we take the VMC estimate, and otherwise take the AVF estimate. This incurs a 2x slow-down in the worst case, while remaining much more efficient in many safety critical domains. Neufeld et al. (2014) give better guarantees when combining stochastic estimators.

## 5 Background and Related Work

**Adversarial examples.** Our work on reliability is in part motivated by research on adversarial examples, which highlights the fact that machine learning systems that perform very well on average

may nonetheless perform extremely poorly on particular *adversarial inputs*, often in surprising ways (Szegedy et al., 2013; Goodfellow et al., 2014). Unlike previous work on adversarial examples in reinforcement learning (Huang et al., 2017; Lin et al., 2017), we do not allow adversaries to generate inputs outside the distribution on which the agent is trained.

Most work on adversarial examples has focused on $L_p$ norm balls in the image domain. Many recent papers have questioned the practical value of norm ball robustness (Gilmer et al., 2018; Engstrom et al., 2017; Schott et al., 2018). However, moving beyond the norm ball specification has been difficult, because for unrestricted adversarial examples, human evaluation is necessary to determine the ground truth labels (see Brown et al. (2018); Song et al. (2018) for work in this direction).

In this context, we believe simulated reinforcement learning environments provide a valuable testbed for researchers interested in adversarial examples - the ground truth is provided by the simulator. We hope the domains and problem formulations presented here drive research on adversarial examples beyond norm balls, and towards training and testing models consistent with global specifications.

**Adversarial approaches for evaluation.** O' Kelly et al. (2018) use the cross-entropy method to efficiently estimate the rate of rare but catastrophic accidents in a simulated driving environment. Concurrent work (to ours) by Webb et al. (2019) uses multi-level splitting, a rare event estimation technique, to estimate the proportion of inputs for which a neural network's output violates a particular property. Our aims share similarities to recent proposals for testing components of autonomous vehicle systems (Shalev-Shwartz et al., 2017; Dreossi et al., 2017; Tian et al., 2018; Pei et al., 2017). We believe these approaches are complementary and should be developed in parallel: these works focus on components-level specifications (e.g. controllers should maintain safe distances) while here, we focus on testing the entire agent end-to-end for system-level specifications.

**Rare event estimation.** We draw upon a rich literature on rare event probability estimation (e.g., Bucklew 2004; Rubino & Tuffin 2009; Rubinstein & Kroese 2017). Importance sampling (IS) is one of the workhorses of this field. The closest methods to our approach are the *adaptive importance sampling methods* where data collected from an initial proposal distribution is iteratively used to adjust it to maximize sampling efficiency (e.g., Rubinstein 1997; Rubinstein & Kroese 2017; Li et al. 2013). These methods do not work well in our context where the failure signal is binary. A key novelty in our method is that we adapt the proposal distribution using data from related, but less robust, agents. Our approach is also different from previous works in that, to better reflect the practicalities of RL tasks, we explicitly separate controllable randomness (i.e., initial conditions, simulator parameters) from randomness that is neither controllable, nor observable (environment and agent randomness). As we show, this has implications on the form of the minimum-variance proposal distribution.

In robotics, the assessment of the safety of motion control algorithms has been recognized as a critical aspect of their real-world deployment (Van Den Berg et al., 2011). In a recent paper, extending Janson et al. (2018), Schmerling & Pavone (2017) proposed to use an adaptive mixture importance sampling algorithm to quantify the collision probability for an LQR controller with EKF state estimation applied to non-linear systems with a full rigid-body collision model. Unlike our method, this work needs an analytic model of the environment.

**Learning highly reliable agents.** While in this work we primarily focus on agent evaluation, rather than training, we note recent work on safe reinforcement learning (see García & Fernández, 2015, for a review). These approaches typically rely on strong knowledge of the transition dynamics (Moldovan & Abbeel, 2012), or assumptions about the accuracy of a learned model (Berkenkamp et al., 2017; Akametalu et al., 2014), and we believe are complementary to adversarial testing, which can be viewed as a mechanism for checking whether such assumptions in fact hold. We provide a longer discussion of this literature in Appendix F.

## 6 CONCLUSION AND FUTURE WORK

In this work, we argued that standard approaches to evaluating RL agents are highly inefficient in detecting rare, catastrophic failures, which can create a false sense of safety. We believe the approach and results here strongly demonstrate that adversarial testing can play an important role in assessing and improving agents, but are only scratching the surface. We hope this work lays the groundwork for future research into evaluating and developing robust, deployable agents.

ACKNOWLEDGMENTS

We are grateful to Alhussein Fawzi, Tengyu Ma, and Percy Liang for their helpful input on this paper, and to Dhruva Tirumala, Josh Merel, Abbas Abdolmaleki, Wojtek Czarnecki, Heinrich Kuttler, Yee Whye Teh, Murray Shanahan, Aman Sinha, and Hongseok Namkoong for insightful discussions.

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

## A  RISK ESTIMATION WITH HEAVY-TAILED LOSSES

In this section, we expand on why empirical estimators of the expected risk using uninformed random sampling will have very high variance for problems with heavy-tailed losses. In statistical learning theory, the objective is to find a model $\theta$ which minimizes the *expected risk*, $J = \mathbb{E}_{x \sim D}[\ell_\theta(x)]$, where $\ell_\theta(x)$ denotes the loss of the model $\theta$ on a data point $x$. For example, in classification, $\ell$ is typically the $0 - 1$ loss, indicating whether $\theta$ makes the correct prediction on $x$. In reinforcement learning, $\ell$ returns the negative discounted return on a rollout from state $x$.

To assess a model $\theta$, since we typically can only sample from $D$, the expected risk can be estimated on a *test set* of $n$ i.i.d. points $x_i \sim D$ using the empirical risk : $\hat{J} = \frac{1}{n} \sum_{i=0}^{n-1} \ell_\theta(x_i)$.

The standard justification for this approach is that with high probability, the empirical risk is close to the true expected risk (Valiant, 1984). If $0 \leq \ell(\cdot) \leq a$, then Hoeffding's inequality yields

$$\mathbb{P}[J \geq \hat{J} + \epsilon] \leq exp(-2n\epsilon^2/a^2)$$

Equivalently, the empirical test risk is within $\epsilon$ of the expected risk with probability $1 - \delta$, provided the number of data points in the test set is at least

$$n \geq \frac{a^2 \log(1/\delta)}{2\epsilon^2}$$

In many commonly studied settings, such as classification with the $0 - 1$ loss, or reinforcement learning with $[0, 1]$-bounded rewards and fixed episode lengths (Tassa et al., 2018; Beattie et al., 2016), these bounds guarantee fairly good efficiency. However, for very heavy-tailed losses, these bounds become very weak. Ensuring a constant additive error requires $n = O(a^2)$, and error up to a constant multiple of $a$ still requires $n = O(a)$ examples. Using the self-driving car example from Section 1, consider a car going on many 1-mile-long trips. Due to the extremely negative rewards associated with crashes, achieving a fixed error bound of $\epsilon$ requires $10^{16}$ more trips than would otherwise be necessary if negative rewards for crashes were bounded to the same range as normal operation.

Intuitively, because very costly events may occur with a small probability, random sampling is unlikely to detect these failure modes unless we use a very large number of random samples. Note that for an agent with probability $p$ of failure, $\lim_{p \to 0+} (1 - p)^{c/p} = e^{-c}$. Thus, for small $p$, even with $1/p$ experiments, there is a $> 35\%$ chance the empirical estimate will detect no failures at all, even though for a sufficiently catastrophic failure, this could dominate the overall expected risk. While the exact bound on the error in the empirical estimate may depend on the particular estimator and concentration inequalities being used (Brownlees et al., 2015; Audibert et al., 2011), any approach relying solely on uninformed random sampling will face similar issues.

## B  PROOFS FOR SECTION 3.2

We start with the proof of Proposition 3.2. Let $U_t = W_t c(X_t, Z_t)$ where $W_t = p_X(X_t)/q(X_t)$. Under the condition that $p_X(x)f(x) = 0$ whenever $q(x) = 0$, $\mathbb{E}[U_t] = p$, hence $\mathbb{E}[\hat{P}] = p$.

Since $U_1, \ldots, U_n$ are independent and identically distributed, $\text{Var}[\hat{P}] = \text{Var}[U_t]/n$ for any $t \in [n]$. Hence, the variance of $\hat{P}$ is minimized when the variance of $U_t$ is minimized. Since $\mathbb{E}[U_t] = p$, this variance is minimized when $\mathbb{E}[U_t^2] = \mathbb{E}[W_t^2 \, c^2(X_t, Z_t)] = \mathbb{E}[W_t^2 \, c(X_t, Z_t)] = \mathbb{E}[(W_t f_*^{1/2}(X_t))^2]$ is minimized, where we used the definition of $f_*$ and that $c$ is binary-valued. By integral Cauchy-Schwartz, this is minimized by the proposal distribution whose density with respect to $P_X$ is proportional to $f_*^{1/2}$, leading to the desired claim.

Notably this differs from the deterministic case, where the variance-minimizing proposal distribution has density proportional to $c(X_t)$ with respect to $P_X$. Characterizing the variance-minimizing proposal distribution for the general stochastic case, when $c$ is not binary-valued, is an interesting open question.

Next, we prove Proposition 3.3, which stated that the rejection sampling procedure that accepts a random instance $X \sim P_X$ with probability $f^{1/2}(X)$ produces a sample from $Q_f$. Let $U$ be

uniformly distributed on the $[0,1]$ interval and independent of $X$. Clearly, the probability that a sample produced by rejection sampling falls into a set $A \subset \mathcal{X}$ is $\mathbb{P}(X \in A | U \leq f^{1/2}(X))$. Now,

$$\mathbb{P}(X \in A | U \leq f^{1/2}(X)) = \frac{\mathbb{P}(X \in A, U \leq f^{1/2}(X))}{\mathbb{P}(U \leq f^{1/2}(X))} = \frac{\int_A \mathbb{P}(U \leq f^{1/2}(x)) P_X(x)}{\mathbb{P}(U \leq f^{1/2}(X))}$$
$$= \frac{\int_A f^{1/2}(x) P_X(x)}{\mathbb{P}(U \leq f^{1/2}(X))}.$$

Since $\mathbb{P}(X \in \cdot | U \leq f^{1/2}(X))$ is a probability measure on $\mathcal{X}$, it follows that the unconditional acceptance probability satisfies $\mathbb{P}(U \leq f^{1/2}(X)) = \int_{\mathcal{X}} f^{1/2}(x) P_X(x)$, thus showing that the distribution of accepted points is $Q_f$ as required.

## C  PSEUDOCODE FOR AVF ADVERSARY

---

**Algorithm 2** AVF-guided Search (AVF adversary)

---

**Input:** Sample size $n$
**repeat**
    Collect random initial conditions $S = \{X_1, \ldots, X_n\}$ where $X_i \sim P_X$, $i \in [n]$
    Select $X = \arg\max_{x \in S} f(x)$
    Run an experiment to generate outcome $C = c(X, Z)$
**until** $C = 1$

---

## D  AGENT TRAINING DETAILS

### D.1  ENVIRONMENTS

For the *Driving* domain, we use the TORCS 3D car racing game (Wymann et al., 2000) with settings corresponding to the "Fast car, no bots" setting in Mnih et al. (2016). At each step, the agent receives an 15-dimensional observation vector summarizing its position, velocity, and the local geometry of the track. The agent receives a reward proportional to its velocity along the center of the track at its current position, while collisions with a wall terminate the episode and provide a large negative reward.

Each problem instance is a track shape, parameterized by a 12-dimensional vector encoding the locations and curvatures of waypoints specifying the track. Problem instances are sampled by randomly sampling a set of waypoints from a fixed distribution, and rejecting any tracks with self-intersections.

For the *Humanoid* domain, we use the Humanoid Stand task from Tassa et al. (2018). At each step, the agent receives a 67-dimensional observation vector summarizing its joint angles and velocities, and the locations of various joints in Cartesian coordinates. The agent receives a reward proportional to its head height. If the head height is below $0.7$m, the episode is terminated and the agent receives a large negative reward.

Each problem instance is defined by an initial standing pose. To define this distribution, we sample $1e5$ random trajectories from a separately trained D4PG agent. We sample a random state from these trajectories, ignoring the first 10 seconds of each trajectory, as well as any trajectories terminating in failure, to ensure all problem instances are feasible. Together, we obtain $6e7$ problem instances, so it is unlikely that any problem instance at train or test time has been previously seen.

### D.2  AGENTS

We now describe the agents we used for evaluation on each task. The focus of this work is not on training agents, and hence we leave the agents fixed for all experiments. However, we did make some modifications to decrease agent failure probabilities, as we are most interested in whether we can design effective adversaries when failures are rare, and thus there is not much learning signal to guide the adversary.

On Driving, we use an asynchronous batched implementation of Advantage Actor-Critic, using a V-trace correction, following Espeholt et al. (2018). We use Population-Based Training, and evolve the learning rate and entropy cost weights Jaderberg et al. (2017), with a population size of 5. Each learner is trained for $1e9$ actor steps, which takes 4 hours distributed over 100 CPU workers and a single GPU learner. Since episodes are at most 3600 steps, this equates to roughly $270e3$ episodes per learner, and $1.3e6$ episodes for the entire population. At test time, we take the most likely predicted action, rather than sampling, as we found it decreased the failure probability by roughly half. Additionally, we used a hand-crafted form of adversarial training by training on a more difficult distribution of track shapes, with sharper turns than the original distribution, since this decreased failure probability roughly 20-fold.

On Humanoid, we use a D4PG agent, using the same hyperparameters as Barth-Maron et al. (2018). The agent is trained for $4e6$ learner steps, which corresponds to between $250e6$ and $300e6$ actor steps, which takes 8 hours distributed over 32 CPU workers and a single GPU learner. Since episodes are at most 1000 steps, this equates to roughly $275e3$ episodes. We use different exploration rates on actors, as in Horgan et al. (2018), with noise drawn from a normal distribution with standard deviation $\sigma$ evenly spaced from $0.0$ to $0.4$. We additionally use demonstrations from the agent described in the previous section, which was used for defining the initial pose distribution, following Vecerík et al. (2017). In particular, we use 1000 demonstration trajectories, and use demonstration data for half of each batch to update both the critic and policy networks. This results in a roughly 4-fold improvement in the failure rate.

# E  EXPERIMENTAL DETAILS

## E.1  AVF DETAILS

When constructing the training datasets, we ignore the beginning of training during which the agent fails very frequently. This amounts to using the last $150,000$ episodes of data on Driving and last $200,000$ on Humanoid. To include information about the training iteration of the agent, we simply include a single real value, the current training iteration divided by the maximum number of training iterations. Similarly, for noise applied to the policy, we include the amount of noise divided by the maximum amount of noise. These are all concatenated before applying the MLP. We train both AVF models to minimize the cross-entropy loss, with the Adam optimizer (Kingma & Ba, 2014), for $20,000$ and $40,000$ iterations on Driving and Humanoid respectively, which requires 4.5 and 50 minutes respectively on a single GPU.

On Driving, the AVF architecture uses a 4-layer MLP with 32 hidden units per layer and a single output, with a sigmoid activation to convert the output to a probability. On Humanoid, since failures of the most robust agents are very rare, we use a simplified Differentiable Neural Dictionary architecture (Pritzel et al., 2017) to more effectively leverage the limited number of positive training examples. In particular, to classify an input $x$, the model first retrieves the $K = 32$ nearest neighbors and their corresponding labels $(x_i, y_i)$ from the training set. The final prediction is then a weighted average $(b + \sum_{i:y_i=1} w_i)/(2b + \sum_i w_i)$, where $b$ is a learned pseudocount, which allows the model to make uncertain predictions when all weights are close to 0. To compute weights $w_i$, each point is embedded by a 1-layer MLP $f$ into 16 dimensions, and the weight of each neighbor is computed $w_i = \kappa(f(x), f(x_i))$, where $\kappa$ is a Gaussian kernel, $\kappa(x, y) = exp(\|x - y\|_2^2 /2)$. We tried different hyperparameters for the number of MLP layers on Driving, and the number of neighbors on Humanoid, and selected based on a held-out test set using data collected during training. In particular, to match real-world situations, we did not select hyperparameters based on either failure search or risk estimation experiments.

## E.2  FAILURE SEARCH

We ran the VMC adversary for $5e6$ experiments for each agent. The expected cost is simply $1/p$, where $p$ is the fraction of experiments resulting in failures. For the AVF adversary, we ran $2e4$ experiments for each agent, since failures are more common, so less experiments are necessary to estimate the failure probabilities precisely. For the PR adversary, the adversary first selected problem instances which caused failures on the actor with the least noise, most recent first, followed by the

| Sample size $n$ | $n_0/10$ Cost | $n_0/10$ Speedup | $n_0$ Cost | $10n_0$ Cost | $10n_0$ Speedup |
|---|---|---|---|---|---|
| Driving | 4/23/61 | 0.18/0.22/0.59 | 3/5/11 | 2/3/6 | 1.1/1.7/2.4 |
| Humanoid | 57/173/220 | 0.16/0.23/0.28 | 19/33/56 | 7/11/25 | 1.8/2.4/3.1 |

Table 2: **Adversary hyperparameter sensitivity analysis.** We show the performance of Algorithm 2 for varying values of the sample size parameter $n$. The middle column shows results for $n = n_0$, the value we used in our experiments, and we compare to a factor of 10 larger and smaller. Costs represent expected number of episodes until a failure, and speedups are relative to when $n = n_0$. As before, each cell shows min, median, and max values across 5 agents. On both domains, applying greater optimization pressure improves results.

actor with the second least noise, and so on. We also tried a version which did not account for the amount of noise and simply ran the most recent failures first, which was slightly worse.

For the failure search experiments on the Humanoid domain, we always evaluate the best version of each agent according to the AVF model selection procedure from Section 4.3. We chose this because we are most interested in whether we can design effective adversaries when failures are rare, and simply choosing the final agent is ineffective, as discussed in Appendix E.4.

We now discuss the sample size parameter $n$ in Algorithm 2. As discussed, using larger values of $n$ applies more selection pressure in determining which problem instances to run experiments on, while using smaller values of $n$ draws samples from a larger fraction of $\mathcal{X}$, and thus provides some robustness to inaccuracies in the learned $f$. Of course, other more sophisticated approaches may be much more effective in producing samples from diverse regions of $\mathcal{X}$, but we only try this very simple approach. For our experiments, we use $n = 1000$ for Driving and $n = 10000$ for Humanoid. We did not perform any hyperparameter selection on $n$, in order to match the real-world setting where the experimenter has only a fixed budget of experiments, and wishes to find a single failure. However, we include additional experiments to study the robustness of the AVF adversary to the choice of $n$. These are summarized in Table 2.

We observe that on both domains, applying greater optimization pressure improves results, over the default choices for $n$ we used. However, relative to the improvements over the VMC adversary, these differences are small, and the AVF adversary is dramatically faster than the VMC adversary for all choices of $n$ we tried.

### E.3    RISK ESTIMATION DETAILS

In our paper, we showed that the AVF estimator can estimate the failure probability of an agent much faster than the VMC estimator. In particular, we showed in both the Driving and Humanoid domains that our estimator requires an order of magnitude less episodes to achieve a 3-approximation at any given confidence level. In general, we may be interested in a $\rho$-approximation, that is we might want the estimates to fall in the range $(p/\rho, p\rho)$ where $p$ is the probability of failure that we wish to measure. One might ask whether our results are sensitive to the choice of $\rho$. In other words, did we select $\rho = 3$ so that our results look good? To address this concern, we include results for lower and higher choices of $\rho$. We see that the AVF estimator performs significantly better than the VMC estimator across choices of $\rho$.

### E.4    MODEL SELECTION

**Measuring ground truth failure probabilities.** In Figure 4, we show the "ground truth" failure probabilities used for computing model robustness in Figure 2. Each bar represents a version of the agent at a different point in training, starting from step $5e5$, and taking a new version every $7e4$ learner steps (so that there are 50 agents in total). Failure probabilities are computed by taking the fraction of experiments resulting in failures, using $160,000$ experiments per agent. As in Figure 2, we show robustness $(1/p)$ rather than failure probabilities, so that it is easier to see differences between the most robust models. Robustness should be interpreted as the expected number of experiments until seeing a failure. Thus, when we report robustness in Figure 2 for averages over multiple agents, we first average the failure probabilities of these agents, before applying the recip-

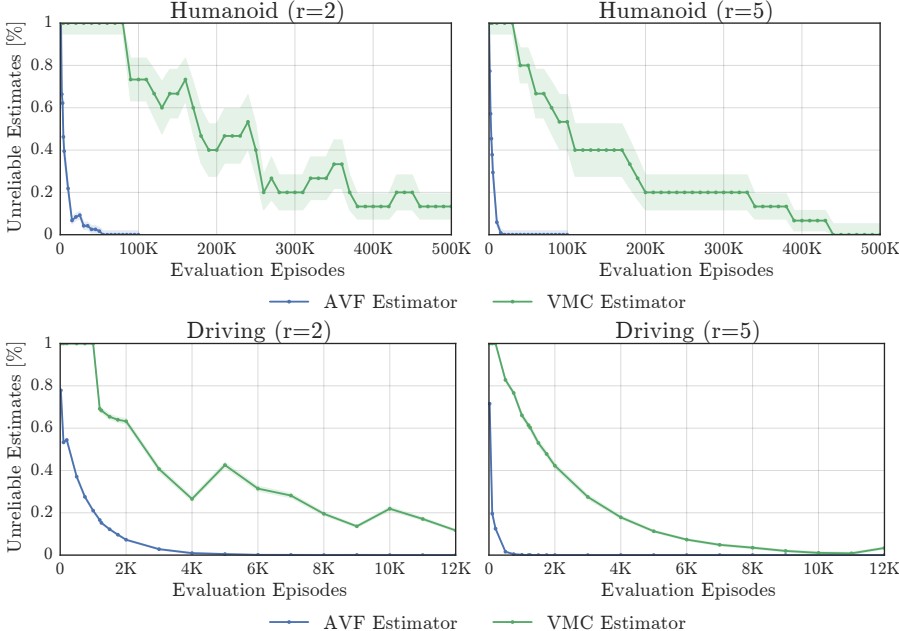

Figure 3: This figure shows the reliability of the AVF and VMC estimators for various choices of $r = \rho$. As before, the $x$ axis shows the number of evaluation episodes. The $y$-axis shows the probability that $\hat{P}$ does not belong in the interval $(p/r, pr)$. The probability is obtained by running the evaluation multiple times, and we include error bars of 2 standard errors. Note that the curves are not smooth especially for the VMC estimator. This is not because of the uncertainty in the plots but is a property of $\rho$-estimators when the sample size is small, that is on the order of $1/p$. For such sample sizes, increasing the sample size does not monotonically improve the estimator's accuracy.

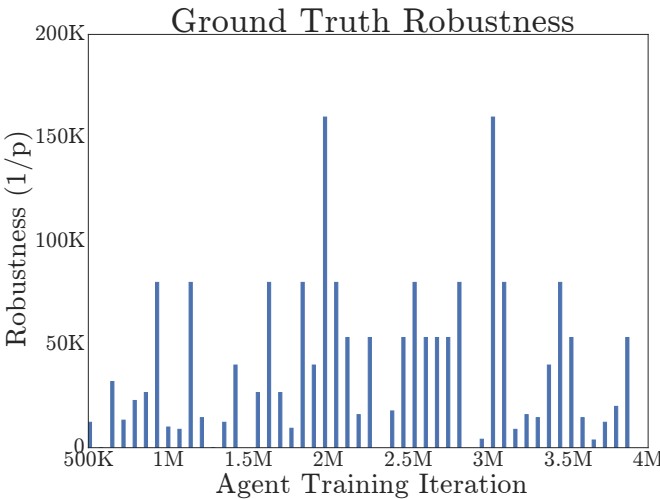

Figure 4: We show the robustness for agents taken from different points in training. The failure probability of each agent is estimated using $160,000$ experiments with VMC. Note that the robustness does not improve monotonically, and that simply choosing the final agent would not provide a robust agent.

rocal, which represents the expected number of experiments until failure, if the agent is randomly selected from that set.

We note that the failure probabilities we report after model selection in Figure 2 are likely conservative. For the two agents which did not fail in our ground truth measurements, we use a failure probability of $1/160,000$ rather than $0$. We could run the ground truth measurements for longer, but already, this required $160,000 * 50 = 8e6$ experiments, or $231$ CPU-days. Nonetheless, even with these estimates, the AVF estimator showed significant improvements over using the VMC estimator for model selection.

**Non-monotonicity of robustness over training.** Finally, we note that the robustness is not monotonically increasing, and thus simply taking the last agent (or randomly choosing from the last few) would be a worse model selection strategy than uniform exploration with the AVF estimator. This differs from the standard setting in deep reinforcement learning, with tightly bounded rewards, where expected return usually (roughly) monotonically increases over the course of training. As discussed in Sections 1 and 5, when the agent is optimized using VMC, failures are observed very rarely, and thus the optimization process does not effectively decrease the failure probability, which is consistent with the non-monotonicity we observe here.

## F    EXTENDED RELATED WORK

**Learning highly reliable agents.** Our results also indicate that learning highly reliable agents will likely require a significant departure from some of current practices. In particular, since for highly reliable systems failure events are extremely rare, any technique that uses vanilla Monte Carlo (that is, perhaps 99% of the current RL literature) is ought to fail to provide the necessary guarantees. This is a problem that has been studied in the simulation optimization literature, but so far it has received only little attention in the RL community (e.g., Frank et al. 2008).

The training time equivalent of our setting can be viewed as a special case of Constrained MDPs (Altman, 1999), where a cost of $1$ is incurred at the final state of a catastrophic trajectory. While recent work has adapted deep reinforcement learning techniques to handle constraints, these address situations where costs are observed frequently, and the objective is to keep expected costs below some threshold (Achiam et al., 2017; Chow et al., 2018) as opposed to avoiding costs entirely, and are not designed to specifically satisfy constraints when violations would be caused by rare events. This suggests a natural extension to the rare failures case using the techniques we develop here. Related work in safe reinforcement learning (see García & Fernández, 2015, for a review) tries to avoid catastrophic failures either during learning or deployment. Approaches for safe exploration typically

rely on strong knowledge of the transition dynamics (Moldovan & Abbeel, 2012), or assumptions about the accuracy of a learned model (Berkenkamp et al., 2017; Akametalu et al., 2014), and we believe are complementary to adversarial testing approaches, which make weak assumptions, but also do not provide guarantees. Finally, there has been significant recent interest in training controllers that are robust to system misspecification by exposing them to pre-specified or adversarially constructed noise applied e.g. to observation or system dynamics (Peng et al., 2017; Pinto et al., 2017; Zhu et al., 2018; Mandlekar et al., 2017).

