# OpenReview forum: "Rigorous Agent Evaluation: An Adversarial Approach to Uncover Catastrophic Failures"
_ICLR.cc/2019/Conference_

### Official Review · AnonReviewer3 · 2018-11-01
**Relevant, convincing experiments with a potential weak point in the method**

**Rating:** 6
**Confidence:** 3

**Review:**

PAPER SUMMARY
-------------

The paper proposes a method for evaluating the failure probability of a learned agent, which is important in safety critical domains.

Using plain Monte Carlo for this evaluation can be too expensive, since discovering a failure probability of epsilon requires on the order of 1/epsilon samples. Therefore the authors propose an adversarial approach, which focuses on scenarios which are difficult for the agent, while still yielding unbiased estimates of failure probabilities.

The key idea of the proposed approach is to learn a failure probability predictor (FPP). This function attempts to predict at which initial states the system will fail. This function is then used in an importance sampling scheme to sample the regions with higher failure probability more often, which leads to higher statistical efficiency.
Finding the FPP is itself a problem which is just as hard as the original problem of estimating the overall failure probability. However, the FPP can be trained using data from different agents, not just the final agent to be evaluated (for instance the data from agent training, containing typically many failure cases). The approach hinges on the assumption that these agents tend to fail in the same states as the final agent, but with higher probability.

The paper shows that the proposed method finds failure cases orders of magnitude faster than standard MC in simulated driving as well as a simulated humanoid task. Since the proposed approach uses data acquired during the training of the agent, it has more information at its disposal than standard MC. However, the paper shows that the proposed method is also orders of magnitudes more efficient than a naive approach using the failure cases during training.


REVIEW SUMMARY
--------------

I believe that this paper addresses an important problem in a novel manner (as far as I can tell) and the experiments are quite convincing.
The main negative point is that I believe that the proposed method has some flaws which may actually decrease statistical efficiency in some cases (please see details below).


DETAILED COMMENTS
-----------------

- It seems to me that a weak point of the method is that it may also severly reduce the efficiency compared to a standard MC method. If the function f underestimates the probability of failure at certain x, it would take a very long time to correct itself because these points would hardly ever be evaluated. It seems that the paper heuristically addresses this to some extent using the exponent alpha of the function. However, I think there should be a more in-depth discussion of this issue. An upper-confidence-bound type of algorithm may be a principled way of addressing this problem.

- The proposed method relies on the ability to initialize the system in any desired state. However, on a physical system, where finding failure cases is particularly important, this is usually not possible. It would be interesting if the paper would discuss how the proposed approach would be used on such real systems.

- On page 6, in the first paragraph, the state is called s instead of x as before. Furthermore, the arguments of f are switched.

---

> ### Author Response · Authors · 2018-11-06
> **Clarification regarding Proposition 3.2**
>
> Thank you for the specific feedback and helpful comments. We wanted to quickly clarify the correctness of Proposition 3.2, since it seemed to be a major point in your review.
>
> > "It seems to me that Proposition 3.2 is wrong. In the proof it is written E[U^2] = E[W^2 c(X,Z)], which is wrong since U^2 = W^2 c^2(X,Z). This means that the proposal distribution Q_f* is not in fact the optimal proposal distribution. This is problematic because the entire approach is justified using this argument."
>
> We believe the proof is correct, but this point is indeed subtle, and we’ll clarify it in the paper. In our case c(X, Z) is a Bernoulli random variable. So c^2(X, Z) = c(X, Z), as c(·, ·) is either 0 or 1 and in both cases the square is the identity. This means E[U^2] = E[W^2 c^2(X,Z)] = E[W^2 c(X,Z)]. In the case where c represents an arbitrary distribution, the optimal proposal distribution is more difficult to compute and is a worthwhile question for future work.
>
> We also note that the standard analysis of the optimal proposal distribution under importance sampling does not account for unobserved stochasticity, which we model in Z. This is why the optimal proposal distribution we derive (for Bernoulli random variables) differs from the standard case.
>
> Please let us know if this addresses your concern.

---

> ### Author Response · Authors · 2018-11-08
> **Statistical efficiency and other technical concerns**
>
> Thank you for taking the time to write very thoughtful comments.
>
> > "I believe that this paper addresses an important problem in a novel manner (as far as I can tell) and the experiments are quite convincing."
>
> It sounds like we’re on the same page regarding the importance of the problem, novelty, and experimental sections. You raised some really good points about the technical section, which we discuss below.
>
> > "The main negative point is that I believe that the proposed method has some flaws which may actually decrease statistical efficiency in some cases... It seems to me that a weak point of the method is that it may also severely reduce the efficiency compared to a standard MC method."
>
> Theoretically, we can ensure that our method never does more than 2x worse than standard MC.
>
> (1) Here’s an intuitive approach for limiting slowdown by a constant factor. We can run both standard MC and our estimator in parallel. If standard MC finds at least a few failures, we can use standard MC. If not, we can use our method. This incurs a slow-down of 2x in the worst case, while remaining orders of magnitudes better in safety critical domains such as the ones we test. Neufeld et al, which we mention in our related works, give even better guarantees when combining stochastic estimators.
>
> (2) Moreover, any method for variance reduction or choosing proposal distributions can be worse in certain cases. This is true for cross-entropy method, subset simulation, control variates, baselines, to name a few. Yet these methods are used in practice, with great success. Requiring 0 slowdown may be too demanding -- we suspect an analogue of the no free lunch theorem might hold -- but we can limit slowdown by a constant factor. We will make all this more clear in the manuscript.
>
> In practice, we employ safeguards to protect us from the issues you describe. (1) For the humanoid experiment we used a Differentiable Neural Dictionary described in Appendix E.1 (Pritzel at al, 2017), this was in D.1 in the original version. A DND is a kNN classifier in feature space, but uses a learned pseudo-count to output higher failure probabilities when the query point is far from training points. Intuitively, the DND model outputs higher failure probabilities for points on which it is uncertain, related to UCB. (2) We trained the FPP on weaker agents. So our method typically over-estimates failure probabilities. (3) Even so, if f underestimates the probability of failure at several points x, it will still typically converge much faster than standard MC. If all x are underestimated by at most a factor of k, then our method slows down on the order of sqrt(k). We show experimentally that our method does orders of magnitude better so this slowdown is not bad.
>
> > "The proposed method relies on the ability to initialize the system in any desired state. However, on a physical system, where finding failure cases is particularly important, this is usually not possible. It would be interesting if the paper would discuss how the proposed approach would be used on such real systems."
>
> Our method actually does not initialize the system at arbitrary states. We only assume that the initial state x is sampled from some (unknown) distribution. Further, the initial system state only needs to be partially observable and the unobserved details can be absorbed into Z. We will make this more clear in the paper - does this address your concern?
>
> > "On page 6, in the first paragraph, the state is called s instead of x as before. Furthermore, the arguments of f are switched."
>
> Thanks for spotting this, we will fix this.
>
> References (also cited in the original paper):
> James Neufeld, Andras Gyorgy, Csaba Szepesvari, Dale Schuurmans. Adaptive Monte Carlo via Bandit Allocation. In ICML 2014.
> Alexander Pritzel, Benigno Uria, Sriram Srinivasan, Adria Puigdomenech, Oriol Vinyals, Demis Hassabis, Daan Wierstra, and Charles Blundell. Neural episodic control. In ICML 2017.

---

### Official Review · AnonReviewer1 · 2018-11-02
**Timely topic, reasonable approach, and good experimental results**

**Rating:** 6
**Confidence:** 3

**Review:**

This paper proposed an adversarial approach to identifying catastrophic failure cases in reinforcement learning. It is a timely topic and may have practical significance. The proposed approach is built on importance sampling for the failure search and function fitting for estimating the failure probabilities. Experiments on two simulated environments show significant gain of the proposed approaches over naive search.

The reviewer is not familiar with this domain, but the baseline, naive search, seems like straightforward and very weak. Are there any other methods for the same problem in the literature? The authors may consider to contrast to them in the experiments.

What is the certainty equivalence approach? A reference would be helpful and improve the presentation quality of the paper.

What is exactly the $\theta_t$ in Section 3.3? What is the dimension of this vector in the experiments? What quantities should be encoded in this vector in practice?

I am still concerned about the fact that the FPP depends on the generalization of the binary classification neural network, although the authors tried to give intuitive examples and discussions. Nonetheless, I understand the difficulty. Could the authors give some conditions under which the approach would fail? Any alternative approaches to the binary neural network? What is a good principle to design the network architecture?

Overall, this paper addresses a practically significant problem and has proposed reasonable approaches. While I still have concerns about the practical performance of the proposed methods, this work along the right track in my opinion.

---

> ### Author Response · Authors · 2018-11-09
> **Addressing main concerns regarding practical performance and baselines**
>
> > Overall, this paper addresses a practically significant problem and has proposed reasonable approaches. While I still have concerns about the practical performance of the proposed methods, this work along the right track in my opinion.
>
> Thank you for the positive comments, and helpful feedback. Could you please explain what concerns you have about the practical performance of the proposed methods? How can we address these? We believe our approach is a large improvement over baselines, both in theory, and as supported by our experiments.
>
> > The reviewer is not familiar with this domain, but the baseline, naive search, seems like straightforward and very weak. Are there any other methods for the same problem in the literature?
>
> We assume you are talking about failure search, and not failure rate estimation? In our original paper, we did compare our method with an additional baseline: a prioritized replay baseline. This does significantly better than naive search, but significantly worse than our proposed method.
>
> We seem to be the first to tackle this problem. The setting is sufficiently different from classical settings, so classical baselines would not work, as we explain in our response to R2. We’d be happy to compare to additional baselines though - are there are any other baselines you would suggest we include?
>
> > I am still concerned about the fact that the FPP depends on the generalization of the binary classification neural network, although the authors tried to give intuitive examples and discussions. Nonetheless, I understand the difficulty. Could the authors give some conditions under which the approach would fail? Any alternative approaches to the binary neural network? What is a good principle to design the network architecture?
>
> The main point we hope to convey is that approaches beyond VMC are crucial, and using an optimized adversary is a good idea in safety-critical settings. We can guarantee that we never do worse than VMC by over a small constant factor (see the discussion on statistical efficiency in our response to R3 for details). However, as you point out, details can influence how much improvement we observe in practice. These details can be application specific, and is not the focus of our paper, but we expand on some of these details below.
>
> Our approach would not help if the neural network severely underestimates the failure probability of a large fraction of failure cases. This could occur for initial states that are very different from all the initial states we have seen during training. We could mitigate this issue: (1) In the humanoid domain, we use a differentiable neural dictionary. The DND outputs higher failure probabilities for points very far from those seen during training. (2) Since we train on weaker agents, we tend to overestimate the failure probabilities. In general, a guiding principle is to output higher failure probabilities for examples we are uncertain about.
>
> We included architectural details in Appendix D.1, but will move the key ideas to the main paper in the next update. Does this address your concerns? We are happy to provide more details if that helps.

---

> > ### Comment · AnonReviewer1 · 2018-11-27
> > **Appreciate the clarifications**
> >
> > Thank the authors for clarifying some of the details.
> >
> > Regarding my concern about the practical performance of the proposed approach, I was not referring to the experiments but rather the use cases in the real world. As FPP depends on the generalization of the binary classification neural network, it is hard to give a confidence interval about its prediction. Furthermore, the distribution of the training data for this binary neural network may be significantly mismatched with that of the test given that the catastrophic failures are rare in nature.
> >
> > I do not have any concrete baselines in mind. I find it hard to justify the proposed approach by experimental comparisons only. Reiterating my earlier comments, it is better to lay out the conditions under which the proposed method would (not) work. The authors' current answer is somehow too obvious to be useful for practitioners: "if the neural network severely underestimates the failure probability of a large fraction of failure cases".

---

> > > ### Author Response · Authors · 2018-11-27
> > > **Real-world applicability**
> > >
> > > Thanks for clarifying your concerns.
> > >
> > > We understand the high-level question raised here to be: “when should practitioners deploying a system in the real world test this system with the FPP rather than VMC”? In short, the answer is *always*.
> > >
> > > First, for risk estimation, by mixing the FPP and VMC estimates, we can guarantee that we never do worse than VMC by over a small constant factor, even when the FPP does not generalize at all, while preserving orders of magnitudes improvement when the FPP generalizes. See the discussion on statistical efficiency in our response to R3, or the paper by Neufeld et al in our citations, for details.
> > >
> > > Second, in the real world, practitioners have limited evaluation budgets - self-driving car companies can’t test the car for millions of miles before every code update. When we deploy ML in safety critical environments, existing methods will often find 0 failures under limited evaluation, even if the system is unsafe. We don’t claim that our method will work well in all such situations. But even if it finds failures in some safety critical domains, preventing the deployment of some unsafe systems is a very positive impact. Our experiments suggest that there are widely used domains where our method works. Further, conceptually, the proposed continuation method for learning the FPP seems much stronger to us than all existing methods and baselines we are aware of.
> > >
> > > In the revised section 3.3, we gave detailed explanations for when the method would (not) be better than existing approaches. To add on, if the test agent fails in a subset of ways (at least some of) the training agents do, our method will work well. If all the training agents do well in a particular scenario, but the test agent fails, then we will do no better (but no worse) than existing methods at detecting such failures.
> > >
> > > Finally, we've also provided a number of tips for practitioners using our method in the revised section 4.4, for example using a DND/Bayesian Neural Network to prioritize sampling uncertain states.

---

> ### Author Response · Authors · 2018-11-09
> **Clarifying other details**
>
> > What is the certainty equivalence approach? A reference would be helpful and improve the presentation quality of the paper.
>
> The certainty equivalence approach is described on page 3. The term has a long history in economics and control, going back to work by Stephen Turnovsky. We will add a reference:
> Stephen Turnovsky. Optimal Stabilization Policies for Stochastic Linear Systems: The Case of Correlated Multiplicative and Additive disturbances. Review of Economic Studies 1976. 43 (1): 191–94.
>
> > What is exactly the $\theta_t$ in Section 3.3? What is the dimension of this vector in the experiments? What quantities should be encoded in this vector in practice?
>
> In general, theta_t should contain any features which provide useful information about the failure probabilities of the policy, and are easy to condition on. In our experiments, theta_t encodes the training iteration, and the amount of noise applied to the policy (details in old appendix D.1, moved to E.1 in the upcoming version), so two dimensions. More features may improve performance, but this was just the simple thing we tried, and since the improvement was already so drastic, it didn’t seem there was much point pushing further.

---

### Official Review · AnonReviewer2 · 2018-11-06
**Effective application of an importance sampling framework to testing RL agent policies for rare failures**

**Rating:** 6
**Confidence:** 3

**Review:**

Summary:
Proposes an importance sampling approach to sampling failure cases for RL algorithms. The proposal distribution is based on a function learned via a neural network on failures that occur during agent training. The method is compared to random sampling on two problems where the "true" failure probability can be approximated through random sampling. The IS method requires substantially fewer samples to produce failure cases and to estimate the failure probability.

Review:
The overall approach is technically sound, and the experiments demonstrate a significant savings in sampling compared to naive random sampling. The specific novelty of the approach seems to be fitting the proposal distribution to failures observed during training.

I think the method accomplishes what it sets out to do. However, as the paper notes, creating robust agents will require a combination of methodologies, of which this testing approach is only a part.

I wonder if learning the proposal distribution based on failures observed during training presents a risk of narrowing the range of possible failures being considered. Of course identifying any failure is valuable, but by biasing the search toward failures that are similar to failures observed in training, might we be decreasing the likelihood of discovering failures that are substantially different from those seen during training? One could imagine that if the agent has not explored some regions of the state space, we would actually like to sample test examples from the unexplored states, which becomes less likely if we preferentially sample in states that were encountered in training.

The paper is well-written with good coverage of related literature. I would suggest incorporating some of the descriptions of the models and methods in Appendix D into the main paper.

Comments / Questions:
* Sec 4.2: How are the confidence bounds for the results calculated?
* What are the "true" failure probabilities in the experiments?
* Sec 4.3: There is a reference to non-existant "Appendix X"

Pros:
* Overall approach is sound and achieves its objectives

Cons:
* Small amount of novelty; primarily an application of established techniques

---

> ### Author Response · Authors · 2018-11-08
> **Addressing main concerns and novelty**
>
> Thank you for the review and suggestions. We first address what we understand to be the main concerns in your review:
>
> We believe there are two sources of novelty. (1) A long-term goal is robust RL agents. Testing agents when rewards are highly sparse is on the critical path to this goal. To our knowledge, this problem has gone unaddressed. Thus, one novelty is considering a practical and important class of rare event estimation problems. (2) Our setting is fairly different from classical settings. By exploiting its structure, we provide an effective approach, whereas prior approaches simply would not work.
>
> > Small amount of novelty; primarily an application of established techniques
> > The specific novelty of the approach seems to be fitting the proposal distribution to failures observed during training.
>
> We believe there are several novel ideas in our approach which are missing in this summary. These novelties aren’t just small changes - we don’t see how existing approaches could handle our setting (failure search and risk estimation, with binary failure signals) without them. Admittedly, we emphasized importance over novelty in writing the paper, and will edit for clarity.
>
> The main novelty in the continuation approach is to learn the proposal distribution from a family of related, but weaker, agents. Our method goes beyond simply fitting a function to data. Fitting a proposal distribution to failures observed for the final agent would not work well. For example, in Humanoid, the final agent fails once every 110k episodes, and was trained for 300k episodes. If we run existing methods like the cross-entropy method on the final agent, we would need significantly more than 300k episodes of data to get a good proposal distribution.
>
> Another novel aspect is our extension of the standard importance sampling setup to include stochasticity. While this seems very fundamental, we are not aware of this in prior work. To reflect the practicalities of RL tasks, we separate controllable randomness (observed initial conditions) from unobservable, uncontrollable randomness (environment and agent randomness, or unobserved initial conditions). We show this changes the form of the minimum-variance proposal distribution (Proposition 3.2). Additionally, in our setup, the initial state distribution is arbitrary and unknown.
>
> > I wonder if learning the proposal distribution based on failures observed during training presents a risk of narrowing the range of possible failures being considered.
>
> This is a good observation. In our humanoid experiments, we safeguard against this using a differentiable neural dictionary (Appendix D.1, moved to E.1 in the latest revision). This encourages higher failure probabilities for initial conditions far from those seen during training. Also see our response to R3 regarding statistical efficiency.

---

> ### Author Response · Authors · 2018-11-08
> **Other details**
>
> > I think the method accomplishes what it sets out to do. However, as the paper notes, creating robust agents will require a combination of methodologies, of which this testing approach is only a part.
>
> Agreed, this an exciting direction for future work. We believe our work is essential for this goal - if we cannot test whether an agent is robust or not, we cannot hope to develop robust agents. Note that in section 4.3 we use the FPP in a simple way to identify more robust agents. We hope future work extends on this - one way is to learn the FPP online with the policy and apply it for adversarial training. This could yield large improvements in sample efficiency - if the FPP is 100x faster at failure search, the agent gets useful examples 100x as often.
>
> > I would suggest incorporating some of the descriptions of the models and methods in Appendix D into the main paper.
>
>
> We’ve edited down the length of the paper, which allows to move some important details to the main paper. We’ll mention some details regarding the training + architecture of the failure probability predictor in the next update. Are there any specific details you would suggest we include?
>
> > Sec 4.2: How are the confidence bounds for the results calculated?
> > What are the "true" failure probabilities in the experiments?
>
> The ground truth failure probabilities are obtained by running the VMC estimator for 5e6 episodes on Driving and 2e7 episodes on Humanoid. Right now, this is mentioned in the footnote at the bottom of page 7, with additional details in the appendix. Thanks for raising this - we’ve definitely tried to make these details as clear as possible, but also realize there’s a lot of such details, and may still be unclear. Please let us know if the writing could be clearer.
>
> The confidence bands in Figure 1 represent 2 standard errors. Each plot is generated by running the estimators many times, and plotting the probability of an unreliable estimate. We use a conservative estimate for standard errors, where if p^ is the empirical mean over n trials for the probability parameter for a Bernoulli RV, SE(p^) = sqrt(max(p^, 0.1) * (1-p^) / n). The max is just to avoid overly narrow confidence bands when p^ is very close to 0 (i.e. when none of the estimates from the estimator are unreliable).
>
> > Sec 4.3: There is a reference to non-existant "Appendix X"
>
> Thanks, fixed.

---

### Author Response · Authors · 2018-11-19
**Paper Updated**

Dear Reviewers,

Thank you for the constructive feedback. All reviews expressed that we were formulating and tackling a significant problem, and that the experimental results were compelling. There were also positive comments about the soundness and novelty of the approach. We hope our work leads to an increased focus on robustness and adversarial examples in RL (and in general beyond norm-ball perturbations).

We have updated our paper to incorporate reviewer feedback. In particular, we added a paragraph at the end of section 4.1 to explain why classical baselines would not work in our context. We added section 4.4 to discuss practical considerations: lower bounds on statistical efficiency, as well as heuristics we use to robustify our method. We have revamped the exposition in section 3.3 to explain one of the key novelties of our approach: the continuation approach to learning FPPs. The other novelties were motivating an important, unaddressed problem, and the extension of the importance sampling framework to include stochasticity.

We believe these address the reviewer comments on statistical efficiency, baselines, and novelty. If the responses satisfy the reviewers, we hope they will consider raising their scores, or letting us know in what ways they think the paper should be improved.

Thanks,
Authors

---

### Meta-Review · Area_Chair1 · 2018-12-15
**reasonable approach, convincing experiments, important topic**

**Confidence:** 3
**Recommendation:** Accept (Poster)

**Metareview:**


* Strengths

The paper addresses a timely topic, and reviewers generally agreed that the approach is reasonable and the experiments are convincing. Reviewers raised a number of specific concerns (which could be addressed in a revised version or future work), described below.

* Weaknesses

Some reviewers were concerned the baselines are weak. Several reviewers were concerned that relying on failures observed during training could create issues by narrowing the proposal distribution (Reviewer 3 characterizes this in a particularly precise manner). In addition, there was a general feeling that more steps are needed before the method can be used in practice (but this could be said of most research).

* Recommendation

All reviewers agreed that the paper should be accepted, although there was also consensus that the paper would benefit from stronger baselines and more close attention to issues that could be caused by an overly narrow proposal distribution. The authors should consider addressing or commenting on these issues in the final version.